# CURIOSITY-DRIVEN EXPERIENCE PRIORITIZATION VIA DENSITY ESTIMATION

## ABSTRACT

In Reinforcement Learning (RL), an agent explores the environment and collects trajectories into the memory buffer for later learning. However, the collected trajectories can easily be imbalanced with respect to the achieved goal states. The problem of learning from imbalanced data is a well-known problem in supervised learning, but has not yet been thoroughly researched in RL. To address this problem, we propose a novel Curiosity-Driven Prioritization (CDP) framework to encourage the agent to over-sample those trajectories that have rare achieved goal states. The CDP framework mimics the human learning process and focuses more on relatively uncommon events. We evaluate our methods using the robotic environment provided by OpenAI Gym. The environment contains six robot manipulation tasks. In our experiments, we combined CDP with Deep Deterministic Policy Gradient (DDPG) with or without Hindsight Experience Replay (HER). The experimental results show that CDP improves both performance and sample-efficiency of reinforcement learning agents, compared to state-of-the-art methods.

## 1 INTRODUCTION

Reinforcement Learning (RL) (Sutton & Barto, 1998) combined with Deep Learning (DL) (Goodfellow et al., 2016) led to great successes in various tasks, such as playing video games (Mnih et al., 2015), challenging the World Go Champion (Silver et al., 2016), and learning autonomously to accomplish different robotic tasks (Ng et al., 2006; Peters & Schaal, 2008; Levine et al., 2016; Chebotar et al., 2017; Andrychowicz et al., 2017).

One of the biggest challenges in RL is to make the agent learn sample-efficiently in applications with sparse rewards. Recent RL algorithms, such as Deep Deterministic Policy Gradient (DDPG) (Lillicrap et al., 2015), enable the agent to learn continuous control, such as manipulation and locomotion. Furthermore, to make the agent learn faster in the sparse reward settings, Andrychowicz et al. (2017) introduced Hindsight Experience Replay (HER) that encourages the agent to learn from whatever goal states it has achieved. The combination use of DDPG and HER lets the agent learn to accomplish more complex robot manipulation tasks. However, there is still a huge gap between the learning efficiency of humans and RL agents. In most cases, an RL agent needs millions of samples before it becomes good at the tasks, while humans only need a few samples (Mnih et al., 2015).

One ability of humans is to learn with curiosity. Imagine a boy learning to play basketball and he attempting to shoot the ball into the hoop. After a day of training, he replayed the memory about the moves he practiced. During his recall, he realized that he missed most of his attempts. However, a few made contact with the hoop. These near successful attempts are more interesting to learn from. He will put more focus on learning from these. This kind of curiosity-driven learning might make the learning process more efficient.

Similar curiosity mechanisms could be beneficial for RL agents. We are interested in the RL tasks, in which the goals can be expressed in states. In this case, the agent can analyze the achieved goals and find out which states have been achieved most of the time and which are rare. Based on the analysis, the agent is able to prioritize the trajectories, of which the achieved goal states are novel. For example, the goal states could be the position and the orientation of the target object. We want to encourage the agent to balance the training samples in the memory buffer. The reason is that the policy of the agent could be biased and focuses on a certain group of achieved goal states. This causes the samples to be imbalanced in the memory buffer, which we refer to as memory imbalance.

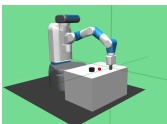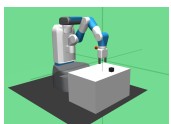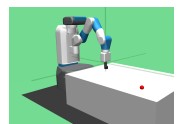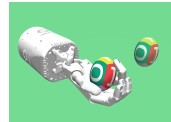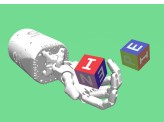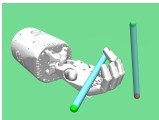

Figure 1: Robot arm Fetch and Shadow Dexterous hand environment: `FetchPush`, `FetchPickAndPlace`, `FetchSlide`, `HandManipulateEgg`, `HandManipulateBlock`, and `HandManipulatePen`.

To overcome the class imbalance issue in supervised learning, such as training deep convolutional neural networks with biased datasets, researchers utilized over-sampling and under-sampling techniques (Deng et al., 2009; Felzenszwalb et al., 2008; Buda et al., 2018; Galar et al., 2012). For instance, the number of one image class is significantly higher than another class. They over-sampled the training images in the smaller class to balance the training set and ultimately to improve the classification accuracy. This idea could be combined with experience replay in RL. We investigate into this research direction and propose a novel curiosity-based prioritization framework for reinforcement learning agents.

In this paper, we introduce a framework called Curiosity-Driven Prioritization (CDP) which allows the agent to realize a curiosity-driven learning ability similar to humans. This approach can be combined with any off-policy RL algorithm. It is applicable whenever the achieved goals can be described with state vectors. The pivotal idea of CDP is to first estimate the density of each achieved goal and then prioritize the trajectories with lower density to balance the samples that the agent learns from. To evaluate CDP, we combine CDP with DDPG and DDPG+HER and test them in the robot manipulation environments.

## 2 BACKGROUND

In this section, we introduce the preliminaries, such as the experiment environments, the reinforcement learning approaches and the density estimation algorithm we used in the experiments.

### 2.1 ENVIRONMENTS

The environment we used in our experiments is the robotic simulations provided by OpenAI Gym (Brockman et al., 2016; Plappert et al., 2018), using the MuJoCo physics engine (Todorov et al., 2012). The robotic environment is based on currently existing robotic hardware and is designed as a standard benchmark for Multi-goal RL. The robot agent is required to complete several tasks with different goals in each scenario. There are two kinds of robot agents in the environment. One is a 7-DOF Fetch robotic arm with a two-finger gripper as an end-effector. The other is a 24-DOF Shadow Dexterous robotic hand. We use six challenging tasks for evaluation, including push, slide, pick & place with the robot arm, and hand manipulation of the block, egg, and pen, see Figure 1.

**Goals:** The goals $g$ are the desired positions and the orientations of the object.

**States:** The system states $s$ in the simulation consist of positions, orientations, linear and angular velocities of all robot joints and of an object. The state $s$ consists of two sub-vectors, the achieved goal state $s^g$ and the context state $s^c$, i.e. $s = (x^g\|x^c)$, where $\|$ denotes concatenation. In our case, the achieved goal state $s^g$ represents the positions and the orientations of the object, which has the same dimension as the real goal $g$. The context state $s^c$ contains the reset system information, including the linear and angular velocities of all robot joints and of an object. The sate input to the universal value function, see Section 2.2, is the system state $s$ combined with the real goal $g$, i.e. $(s\|g)$.

**Rewards:** In all environments, we consider sparse rewards $r$. There is a tolerant range between the desired goal states and the achieved goal states. If the object is not in the tolerant range of the real goal, the agent receives a reward signal -1 for each transition; otherwise, the reward signal is 0.

## 2.2 Reinforcement Learning

**Markov Decision Process:** We consider an agent interacting with an environment. We assume the environment is fully observable, including a set of state $\mathcal{S}$, a set of action $\mathcal{A}$, a distribution of initial states $p(s_0)$, transition probabilities $p(s_{t+1}|s_t, a_t)$, a reward function $r \colon \mathcal{S} \times \mathcal{A} \to \mathbb{R}$, and also a discount factor $\gamma \in [0, 1]$. These components formulate a Markov decision process represented as a tuple, $(\mathcal{S}, \mathcal{A}, p, r, \gamma)$. A policy $\pi$ maps a state to an action, $\pi \colon \mathcal{S} \to \mathcal{A}$.

**Deep Deterministic Policy Gradient:** The objective, expected return $\mathbb{E}_{s_0}[R_0|s_0]$, can be maximized using temporal difference learning, policy gradients, or the combination of both, i.e. the actor-critic methods (Sutton & Barto, 1998). For continuous control tasks, Deep Deterministic Policy Gradient (DDPG) shows promising performance, which is essentially an off-policy actor-critic method (Lillicrap et al., 2015).

**Universal Value Function Approximators:** For multi-goal continuous control tasks, DDPG can be extended with Universal Value Function Approximators (UVFA) (Schaul et al., 2015a). UVFA essentially generalizes the Q-function to multiple goal states $g \in \mathcal{G}$. Now, the Q-value depends not only on the state-action pairs, but also depends on the goals: $Q^{\pi}(s_t, a_t, g) = \mathbb{E}[R_t|s_t, a_t, g]$.

**Hindsight Experience Replay:** For robotic tasks, if the goal is challenging and the reward is sparse, then the agent could perform badly for a long time before learning anything. Hindsight Experience Replay (HER) encourages the agent to learn from whatever goal states that it has achieved. Andrychowicz et al. (2017) show that HER makes training possible in challenging robotic environments. However, the episodes are uniformly sampled in the replay buffer, and subsequently, the virtual goals are sampled from the episodes. More sophisticated replay strategies are requested for improving sample-efficiency (Plappert et al., 2018).

## 2.3 Density Estimation Methods

For estimating the density $\rho$ of the achieved goals in the memory buffer, we use a Gaussian mixture model because it can be trained reasonably fast for RL agents. GMM is also much faster in inference compared to Kernel Density Estimate (KDE) (Rosenblatt, 1956). Gaussian Mixture Model (GMM) (Duda & Hart, 1973; Murphy, 2012) is a probabilistic model that assumes all the data points are generated from $K$ Gaussian distributions with unknown parameters, mathematically: $\rho(\mathbf{x}) = \sum_{k=1}^{K} c_k \mathcal{N}(\mathbf{x}|\boldsymbol{\mu}_k, \boldsymbol{\Sigma}_k)$. Every Gaussian density $\mathcal{N}(\mathbf{x}|\boldsymbol{\mu}_k, \boldsymbol{\Sigma}_k)$ is a component of the GMM and has its own mean $\boldsymbol{\mu}_k$ and covariance $\boldsymbol{\Sigma}_k$. The parameters $c_k$ are the mixing coefficients. In our experiments, we use Variational Gaussian Mixture Model (V-GMM) (Blei et al., 2006). The reason is that V-GMM has a natural tendency to set some mixing coefficients $c_k$ close to zero and generalizes better. Therefore, we decide to use V-GMM in our framework as a proof of concept.

## 3 Method

In this section, we formally describe our method, including the motivation, the framework, a mathematical grounding, and a comparison with prioritized experience replay (Schaul et al., 2015b).

### 3.1 Motivation

The motivation of incorporating curiosity mechanisms into RL agents is motivated by the human brain. Recent neuroscience research (Gruber et al., 2014) has shown that curiosity can enhance learning. They discovered that when curiosity motivated learning was activated, there was increased activity in the hippocampus, a brain region that is important for human memory. To learn a new skill, such as playing basketball, people practice repeatedly in a trial-and-error fashion. During memory replay, people are more curious about the episodes that are relatively different and focus more on those. This curiosity mechanism has been shown to speed up learning.

Secondly, the inspiration of how to design the curiosity mechanism for RL agents comes from the supervised learning community, in particular the class imbalance dataset problem. Real-world datasets commonly show the particularity to have certain classes to be under-represented compared to other classes. When presented with complex imbalanced datasets, standard learning algorithms, including neural networks, fail to properly represent the distributive characteristics of the data and thus

provide unfavorable accuracies across the different classes of the data (He & Garcia, 2008; Galar et al., 2012). One of the effective methods to handle this problem is to over-sample the samples in the under-represented class. Therefore, we prioritize the under-represented trajectories with respect to the achieved goals in the agent's memory buffer to improve the performance.

## 3.2 CURIOSITY-DRIVEN PRIORITIZATION

In this section, we formally describe the Curiosity-Driven Prioritization (CDP) framework. In a nutshell, we first estimate the density of each trajectory according to its achieved goal states, then prioritize the trajectories with lower density for replay.

### 3.2.1 COLLECTING EXPERIENCE

At the beginning of each episode, the agent uses partially random policies, such as $\epsilon$-greedy, to start to explore the environment and stores the sampled trajectories into a memory buffer for later replay.

A complete trajectory $\tau$ in an episode is represented as a tuple $(\mathcal{S}, \mathcal{A}, p, r, \gamma)$. A trajectory contains a series of continuous states $s_t$, see Section 2.1, where $t$ is the timestep $t \in \{0, 1, .., T\}$. Each state $s_t \in \mathcal{S}$ also includes the state of the achieved goal $s_t^g$. The density of a trajectory, $\rho$, only depends on the goal states, $s_0^g, s_1^g, ..., s_T^g$.

### 3.2.2 DENSITY ESTIMATION

After the agent collected a number of trajectories, we can fit the density model. The density model we use here is the Variational Gaussian Mixture Model (V-GMM) as introduced in Section 2.3. The V-GMM fits on the data in the memory buffer every epoch and refreshes the density for each trajectory in the buffer. During each epoch, when the new trajectory comes in, the density model predicts the density $\rho$ based on the achieved goals of the trajectory as:

$$\rho = \text{V-GMM}(\tau) = \sum_{k=1}^{K} c_k \mathcal{N}(\tau|\boldsymbol{\mu}_k, \boldsymbol{\Sigma}_k) \tag{1}$$

where $\tau = (s_0^g \| s_1^g \| ... \| s_T^g)$ and each trajectory $\tau$ has the same length. We normalize the trajectory densities using

$$\rho_i = \frac{\rho_i}{\sum_{n=1}^{N} \rho_n} \tag{2}$$

where $N$ is the number of trajectories in the memory buffer. Now the density $\rho$ is between zero and one, i.e. $0 \le \rho \le 1$, After calculating the trajectory density, the agent stores the density value along with the trajectory in the memory buffer for later prioritization.

### 3.2.3 PRIORITIZATION

During replay, the agent puts more focus on the under-represented achieved states and prioritizes the according trajectories. These under-represented achieved goal states have lower trajectory density. We defined the complementary trajectory density as:

$$\bar{\rho} \propto 1 - \rho. \tag{3}$$

When the agent replays the samples, it first ranks all the trajectories with respect to their complementary density values $\bar{\rho}$, and then uses the ranking number (starting from zero) directly as the probability for sampling. This means that the low-density trajectories have high ranking numbers, and equivalently, have higher priorities to be replayed. Here we use the ranking instead of the density directly. The reason is that the rank-based variant is more robust because it is not affected by outliers nor by density magnitudes. Furthermore, its heavy-tail property also guarantees that samples will be diverse (Schaul et al., 2015b). Mathematically, the probability of a trajectory to be replayed after the prioritization is:

$$p(\tau_i) = \frac{\text{rank}(\bar{\rho}(\tau_i))}{\sum_{n=1}^{N} \text{rank}((\bar{\rho}(\tau_n))} \tag{4}$$

where $N$ is the total number of trajectories in the buffer, and $\text{rank}(\cdot) \in \{0, 1, ..., N-1\}$.

### 3.2.4 COMPLETE ALGORITHM

We summarize the complete training algorithm in Algorithm 1.

---

**Algorithm 1** Curiosity-Driven Prioritization (CDP)

---

**Given:**
- an off-policy RL algorithm $\mathbb{A}$         ▷ e.g. DDPG, DDPG+HER
- a reward function $r : \mathcal{S} \times \mathcal{A} \times \mathcal{G} \to \mathbb{R}$.     ▷ e.g. $r(s, a, g) = -1$ (fail), 0 (success)

Initialize neural networks of $\mathbb{A}$, density model V-GMM, and replay buffer $R$
**for** epoch $= 1, M$ **do**
    **for** episode $= 1, N$ **do**
        Sample a goal $g$ and an initial state $s_0$.
        Sample a trajectory $\tau = (s_t \| g, a_t, r_t, s_{t+1} \| g)_{t=0}^T$ using $\pi_b$ from $\mathbb{A}$
        Calculate the densities $\rho$ and $\bar{\rho}$ using Equation (1), (2) and (3)      ▷ estimate density
        Calculate the priority $p(\tau)$ using Equation (4)
        Store transitions $(s_t \| g, a_t, r_t, s_{t+1} \| g, p, \bar{\rho})_{t=0}^T$ in $R$
        Sample trajectory $\tau$ from $R$ based on the priority, $p(\tau)$      ▷ prioritization
        Sample transitions $(s_t, a_t, s_{t+1})$ from $\tau$
        Sample virtual goals $g' \in \{s_{t+1}, ..., s_{T-1}\}$ at a future timestep in $\tau$
        $r'_t := r(s_t, a_t, g')$      ▷ recalculate reward (HER)
        Store the transition $(s_t \| g', a_t, r'_t, s_{t+1} \| g', p, \bar{\rho})$ in $R$
        Perform one step of optimization using $\mathbb{A}$
    **end for**
    Train the density model using the collected trajectories in $R$      ▷ fit density model
    Update the density in $R$ using the trained model      ▷ refresh density
**end for**

---

### 3.3 AN IMPORTANCE SAMPLING PERSPECTIVE

The mathematical explanation for the efficiency of CDP is based on importance sampling. Importance sampling is a general technique to estimate an integral $\int f(x)p(x)dx$ of a function $f(x)$, with the exact distribution $p(x)$ (Murphy, 2012; Owen, 2013). Here, we consider using importance sampling to estimate the integral of the loss function $\mathcal{L}(\tau)$ of the reinforcement learning agent:

$$I = \mathbb{E}[\mathcal{L}] = \int \mathcal{L}(\tau) \frac{p(\tau)}{q(\tau)} q(\tau) d\tau \approx \frac{1}{N} \sum_{i=1}^N \omega_i f(\tau_i) = \hat{I},$$

where $\tau$ is a trajectory, $q(\tau)$ is a proposal distribution, and $\omega_i = p(\tau_i)/q(\tau_i)$ is an importance weight.

The idea here is to draw samples $\tau$ from the buffer in regions which have a high probability, $p(\tau)$, but also where $\mathcal{L}|(\tau)|$ is large. Since, $p(\tau)$ is a uniform distribution, i.e. the agent replays trajectories at random, we only need to draw samples which has large errors $\mathcal{L}|(\tau)|$. The result can be *highly efficient*, meaning the agent needs less samples than sampling from the uniform distribution $p(\tau)$. The CDP framework finds the samples that have large errors based on the 'surprise' of the trajectory. The variance of the estimate $\hat{I}$ is: $\text{var}_q[\mathcal{L}(\tau)\omega(\tau)] = \mathbb{E}_q[\mathcal{L}^2(\tau)\omega^2(\tau)] - I^2$. Since the last term is independent of $q$, we can ignore it. Using Jensen's inequality, we have the following lower bound:

$$\mathbb{E}_q[\mathcal{L}^2(\tau)\omega^2(\tau)] \geqslant (\mathbb{E}_q[|\mathcal{L}(\tau)\omega(\tau)|])^2.$$

To reduce the variance, we set the importance weight as a constant $\omega(\tau) = 1$. This bias-variance trade-off also saves computational time and does not lead to instabilities in our experiment. With CDP, the agent estimates the loss function more efficiently and therefore learns faster. Any density estimation method that can approximate the trajectory density can provide a more efficient proposal distribution $q(\tau)$ than the uniform distribution $p(\tau)$.

### 3.4 COMPARISON WITH PRIORITIZED EXPERIENCE REPLAY

To the best our knowledge, the most similar method to CDP is Prioritized Experience Replay (PER) (Schaul et al., 2015b). To combine PER with HER, we calculate the TD-error of each transition

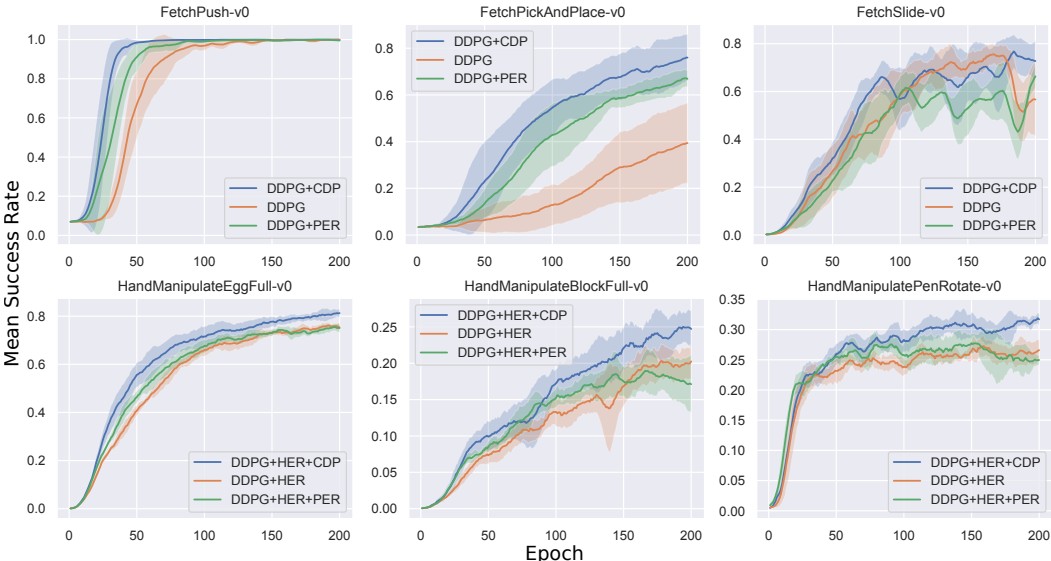

Figure 2: Mean test success rate with standard deviation in all six robot environments

based on the randomly selected achieved goals. Then we prioritize the transitions with higher TD-errors for replay. It is known that PER can become very expensive in computational time (Schaul et al., 2015b), especially when the memory size $N$ is very large. The reason is that PER uses TD-errors for prioritization. After each update of the model, the agent needs to update the priorities of the transitions in the replay buffer, which is $O(\log N)$. In our experiments, see Section 4, we use the efficient implementation based on the "sum-tree" data structure, which can be relatively efficiently updated and sampled from (Schaul et al., 2015b).

Compared to PER, CDP is much faster in computational time because it only updates the trajectory density once per epoch. Due to this reason, CDP is much more efficient than PER in computational time and can be easily combined with any multi-goal RL methods, such as DDPG and HER. In the experiments, Section 4, we first compare the performance improvement of CDP and PER. Afterwards, we compare the time-complexity of PER and CDP. We show that CDP improves performance with much less computational time than PER. Furthermore, the motivations of PER and CDP are different. The former uses TD-errors, while the latter is based on the density of the trajectories.

## 4 EXPERIMENTS

In this section, we investigate the following questions:
 - Does incorporating CDP bring benefits to DDPG or DDPG+HER?
 - Does CDP improve the sample-efficiency in robotic manipulation tasks?
 - How does the density $\bar{\rho}$ relate to the TD-errors of the trajectory during training?

**Performance:** To test the performance difference among DDPG, DDPG+PER, and DDPG+CDP, we run the experiment in the three robot arm environments. We use the DDPG as the baseline here because the robot arm environment is relatively simple. In the more challenging robot hand environments, we use DDPG+HER as the baseline method and test the performance among DDPG+HER, DDPG+HER+PER, and DDPG+HER+CDP.

We compare the mean success rates. Each experiment is carried out across 5 random seeds and the shaded area represents the standard deviation. The learning curve with respect to training epochs is shown in Figure 2. For all experiments, we use 19 CPUs and train the agent for 200 epochs. After training, we use the best-learned policy as the final policy and test it in the environment. The testing results are the final mean success rates. A comparison of the final performances along with the training time is shown in Table 1.

Table 1: Final mean success rate (%) and the training time (hour) for all six environments

| Method | Push | | Pick & Place | | Slide | |
|---|---|---|---|---|---|---|
| | success | time | success | time | success | time |
| DDPG | 99.90% | 5.52h | 39.34% | 5.61h | 75.67% | 5.47h |
| DDPG+PER | 99.94% | 30.66h | 67.19% | 25.73h | 66.33% | 25.85h |
| DDPG+CDP | **99.96%** | 6.76h | **76.02%** | 6.92h | **76.77%** | 6.66h |

| Method | Egg | | Block | | Pen | |
|---|---|---|---|---|---|---|
| | success | time | success | time | success | time |
| DDPG+HER | 76.19% | 7.33h | 20.32% | 8.47h | 27.28% | 7.55h |
| DDPG+HER+PER | 75.46% | 79.86h | 18.95% | 80.72h | 27.74% | 81.17h |
| DDPG+HER+CDP | **81.30%** | 17.00h | **25.00%** | 19.88h | **31.88%** | 25.36h |

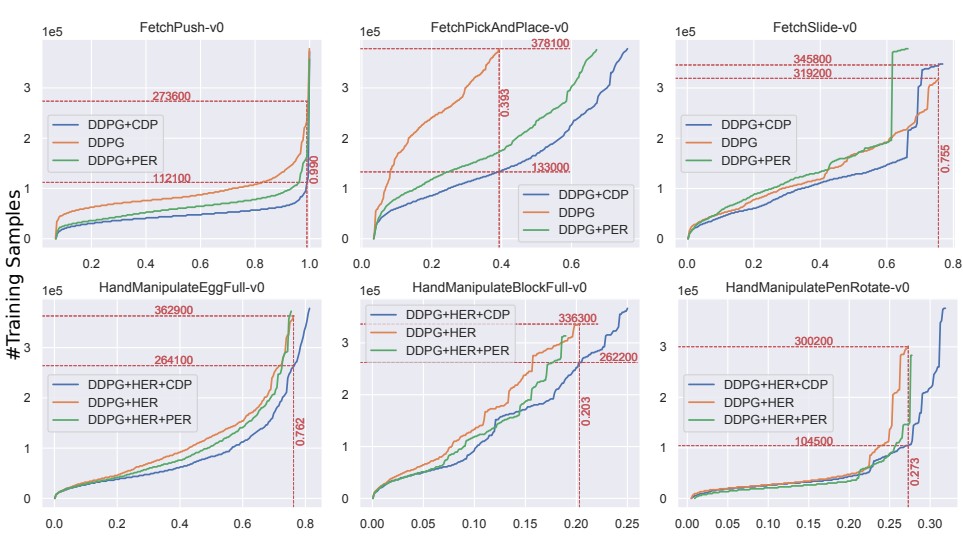

Figure 3: Number of training samples needed with respect to mean test success rate for all six environments (the lower the better)

From Figure 2, we can see that CDP converges faster in all six tasks than both the baseline and PER. The agent trained with CDP also shows a better performance at the end of the training, as shown in Table 1. In Table 1, we can see that the training time of CDP lies in between the baseline and PER. To be more specific, CDP consumes much less computational time than PER does. For example in the robot arm environments, on average DDPG+CDP consumes about 1.2 times the training time of DDPG. In comparison, DDPG+PER consumes about 5 times the training time as DDPG does. In this case, CDP is 4 times faster than PER.

Table 1 shows that baseline methods with CDP give a better performance in all six tasks. The improvement goes up to 39.34 percentage points compared to the baseline methods. The average improvement over the six tasks is 9.15 percentage points. We can see that CDP is a simple yet effective method, improves state-of-the-art methods.

**Sample-Efficiency:** To compare the sample-efficiency of the baseline and CDP, we compare the number of training samples needed for a certain mean test success rate. The comparison is shown in Figure 3. From Figure 3, in the `FetchPush-v0` environment, we can see that for the same 99% mean test success rate, the baseline DDPG needs 273,600 samples for training, while DDPG+CDP only needs 112,100 samples. In this case, DDPG+CDP is more than twice (2.44) as sample-efficient as DDPG. Similarly, in the other five environments, CDP improves sample-efficiency by factors of 2.84, 0.92, 1.37, 1,28 and 2.87, respectively. In conclusion, for all six environments, CDP is able to improve sample-efficiency by an average factor of two (1.95) over the baseline's sample-efficiency.

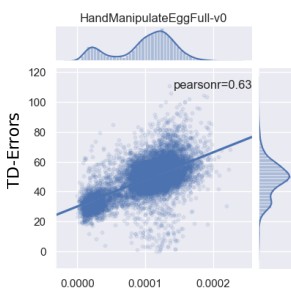 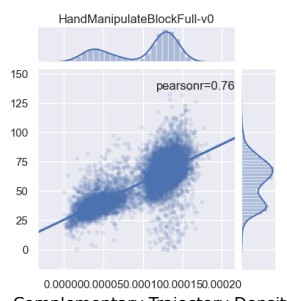 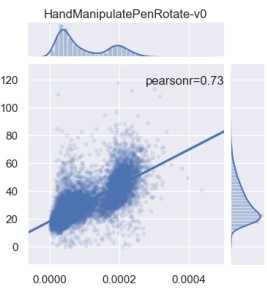

Figure 4: Pearson correlation between the density $\bar{\rho}$ and TD-errors in the middle of training

**Insights:** We also investigate the correlation between the complementary trajectory density $\bar{\rho}$ and the TD-errors of the trajectory. The Pearson correlation coefficient, i.e. Pearson's r (Benesty et al., 2009), between the density $\bar{\rho}$ and the TD-errors of the trajectory is shown in Figure 4. The value of Pearson's r is between 1 and -1, where 1 is total positive linear correlation, 0 is no linear correlation, -1 is total negative linear correlation. In Figure 4, we can see that the complementary trajectory density is correlated with the TD-errors of the trajectory with an average Pearson's r of 0.7. This proves that the relatively rare trajectories in the memory buffer are more valuable for learning. Therefore, it is helpful to prioritize the trajectories with lower density during training.

## 5 RELATED WORK

Experience replay was proposed by Lin (1992) and became popular due to the success of DQN (Mnih et al., 2015). In the same year, prioritized experience replay was introduced by Schaul et al. (2015b) as an improvement of the experience replay in DQN. It prioritized the transitions with higher TD-error in the replay buffer to speed up training. Schaul et al. (2015a) also proposed universal function approximators, generalizing not just over states but also over goals. There are also many other research works about multi-task RL (Schmidhuber & Huber, 1990; Caruana, 1998; Da Silva et al., 2012; Kober et al., 2012; Pinto & Gupta, 2017; Foster & Dayan, 2002; Sutton et al., 2011). Hindsight experience replay (Andrychowicz et al., 2017) is a kind of goal-conditioned RL that substitutes any achieved goals as real goals to encourage the agent to learn something instead of nothing.

Curiosity-driven exploration is a well-studied topic in reinforcement learning (Oudeyer & Kaplan, 2009; Oudeyer et al., 2007; Schmidhuber, 1991; 2010; Sun et al., 2011). Pathak et al. (2017) encourage the agent to explore states with high prediction error. The agents are also encouraged to explore "novel" or uncertain states (Bellemare et al., 2016; Lopes et al., 2012; Poupart et al., 2006; Houthooft et al., 2016; Mohamed & Rezende, 2015; Chentanez et al., 2005; Stadie et al., 2015).

However, we integrate curiosity into prioritization and tackle the problem of data imbalance (Galar et al., 2012) in the memory buffer of RL agents. A recent work (Narasimhan et al., 2015) introduced a form of re-sampling for RL agents based on positive and negative rewards. The idea of our method is complementary and can be combined. The motivation of our method is from the curiosity mechanism in the human brain (Gruber et al., 2014). The essence of our method is to assign priority to the achieved trajectories with lower density, which are relatively more valuable to learn from. In supervised learning, similar tricks are used to mitigate the class imbalance challenge, such as over-sampling the data in the under-represented class (Hinton, 2007; He & Garcia, 2008).

## 6 CONCLUSION

In conclusion, we proposed a simple yet effective curiosity-driven approach to prioritize agent's experience based on the trajectory density. Curiosity-Driven Prioritization shows promising experimental results in all six challenging robotic manipulation tasks. This method can be combined with any off-policy RL methods, such as DDPG and DDPG+HER. We integrated the curiosity mechanism via density estimation into the modern RL paradigm and improved sample-efficiency by a factor of two and the final performance by nine percentage points on top of state-of-the-art methods.

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
