# OpenReview forum: "Curiosity-Driven Experience Prioritization via Density Estimation"
_ICLR.cc/2019/Conference_

### Official Review · AnonReviewer3 · 2018-11-02
**The method proposed in the paper may have poor generalization and scaling performance**

**Rating:** 6
**Confidence:** 4

**Review:**

This work considers a version of importance sampling of states from the replay buffer.  Each trajectory is assigned a rank, inversely proportional to its probability according to a GMM. The trajectories with lower rank are preferred at sampling.

Main issues:

1. Estimating rank from a density estimator

- the reasoning behind picking VGMM as the density estimator is not fully convincing and (dis)advantages of other candidate density estimators are almost not highlighted.

- it is unclear and possibly could be better explained why one needs to concatenate the goals (what would change if we would not concatenate but estimate state densities rather than trajectories?)

2. Generalization issues

- the method is not applicable to episodes of different length
- the approach assumes existence of a state to goal function f(s)->g
- although the paper does not expose this point (it is discussed the HER paper)

3. Scaling issues

- length of the vector grows linearly with the episode length
- length of the vector grows linearly with the size of the goal vector

For long episodes or episodes with large goal vectors it is quite possible that there will not be enough data to fit the GMM model or one would need to collect many samples prior.

4. Minor issues

- 3.3 "It is known that PER can become very expensive in computational time" - please supply a reference


- 3.3 "After each update of the model, the agent needs to update the priorities of the transitions in the replay buffer with the new TD-errors" - However the method only renews priorities of randomly selected transitions (why would there be a large overhead?). Here is from the PER paper "Our final implementation for rank-based prioritization produced an additional 2%-4% increase in running time and negligible additional memory usage"

---

> ### Author Response · Authors · 2018-11-17
> **To Reviewer**
>
> Thank you for the valuable feedback!
> We uploaded a revised version of the paper based on the comments.
>
> - The reason behind using V-GMM is that V-GMM is much faster than KDE in inference and has a better generalization ability compared to GMM. We use V-GMM as a proof of concept for the idea “Curiosity-Driven Experience Prioritization via Density Estimation”. Other density estimation methods can also be applied. We now clarify these reasons in Section “2.3 Density Estimation Methods” of the revised paper.
>
> - We concatenate the goals and estimate the trajectory density instead of state density because HER needs to sample a future state in the trajectory as a virtual goal for training.
>
> - For episodes of different length, we can pad or truncate the trajectories into same lengths and apply V-GMM. Another method is to use PCA or auto-encoder to reduce the dimension into a fixed size and then apply CDP.
>
> - Similarly, to handle scaling issues, for very high dimension vectors, we can first apply dimension reduction methods, such as PCA and auto-encoder, and then use CDP.
>
> - The reference for "It is known that PER can become very expensive in computational time” is actually the “Prioritized Experience Replay” paper itself. On page three of the PER paper, it writes “Implementation: To scale to large memory sizes N , we use a binary heap data structure for the priority queue, for which finding the maximum priority transition when sampling is O(1) and updating priorities (with the new TD-error after a learning step) is O(log N). See Appendix B.2.1 for more details. “
>
> In their Atari case, the memory size N is of 1e4 transitions.
> In our hand manipulation environment cases, the memory size N is of 1e6 trajectories, and each trajectory has 100 transitions. Thus, the memory size is 1e4 (theirs) vs 1e8 (ours). The complexity of updating priorities is O(log N). Therefore, PER is very expensive in computational time, at least in our case.
>
> The memory buffer size N can be found in OpenAI Baselines link: https://github.com/openai/baselines

---

### Official Review · AnonReviewer1 · 2018-11-03
**Handling unbalanced target distributions when conditioning on goal in RL**

**Rating:** 4
**Confidence:** 4

**Review:**

This paper addresses a problem that arises in "universal" value-function approximation (that is, reinforcement-learning when a current goal is included as part of the input);  when doing experience replay, the experience buffer might have much more representation of some goals than others, and it's important to keep the training appropriately balanced over goals.

So, the idea is to a kind of importance weighting of the trajectory memory, by doing a density estimation on the goal distribution represented in the memory and then sample them for training in a way that is inversely related to their densities.  This method results in a moderate improvement in the effectiveness of DDPG, compared to the previous method for hindsight experience replay.

The idea is intuitively sensible, but I believe this paper falls short of being ready for publication for three major reasons.

First, the mechanism provided has no mathematical justification--it seems fairly arbitrary.   Even if it's not possible to prove something about this strategy, it would be useful to just state a desirable property that the sampling mechanism should have and then argue informally that this mechanism has that property.  As it is, it's just one point in a large space of possible mechanisms.

I have a substantial concern that this method might end up assigning a high likelihood of resampling trajectories where something unusual happened, not because of the agent's actions, but because of the world having made a very unusual stochastic transition.   If that's true, then this distribution would be very bad for training a value function, which is supposed to involve an expectation over "nature"'s choices in the MDP.

Second, the experiments are (as I understand it, but I may be wrong) in deterministic domains, which definitely does not constitute a general test of a proposed RL  method.
- I'm not sure we can conclude much from the results on fetchSlide (and it would make sense not to use the last set of parameters but the best one encountered during training)
- What implementation of the other algorithms did you use?

Third, the writing in the paper has some significant lapses in clarity.  I was a substantial way through the paper before understanding exactly what the set-up was;  in particular, exactly what "state" meant was not clear.  I would suggest saying something like s = ((x^g, x^c), g) where s is a state from the perspective of value iteration, (x^g, x^c) is a state of the system, which is a vector of values divided into two sub-vectors, x^g is the part of the system state that involves the state variables that are specified in the goal, x^c (for 'context') is the rest of the system state, and g is the goal.  The dimensions of x^g and g should line up.
- This sentence  was particularly troublesome:  "Each  state s_t also includes the state of the achieved goal, meaning the goal state is a subset of the normal state.  Here, we overwrite the notation s_t  as the achieved goal state, i.e., the state of the object."
- Also, it's important to say what the goal actually is, since it doesn't make sense for it to be a point in a continuous space.  (You do say this later, but it would be helpful to the reader to say it earlier.)

---

> ### Author Response · Authors · 2018-11-17
> **To Reviewer**
>
> Thank you for the valuable feedback!
> We uploaded a revised version of the paper based on the comments.
>
> - We added a mathematical justification paragraph in Section 3.3 “An Importance Sampling Perspective”. We argue that to estimate the integral of the loss function L(τ) of the RL agent efficiently, we need to draw samples τ from the buffer in regions which have a high probability, p(τ), but also where L|(τ)| is large. Since, p(τ) is a uniform distribution, i.e., the agent replays trajectories at random, we only need to draw samples which has large errors L|(τ)|. The result can be highly efficient, meaning the agent needs less samples than sampling from the uniform distribution p(τ). The CDP framework finds the samples that have large errors based on the ‘surprise’ of the trajectory.
>
> Any density estimation method that can approximate the trajectory density can provide a more efficient proposal distribution q(τ) than the uniform distribution p(τ). The sampling mechanism should have a property of oversampling trajectories with larger errors/‘surprise’.
>
> - To mitigate the influence of very unusual stochastic transitions, we use the ranking instead of the density directly. The reason is that the rank-based variant is more robust because it is not affected by outliers nor by density magnitudes. Furthermore, its heavy-tail property also guarantees that samples will be diverse (Schaul et al., 2015b).
>
> - Yes, the experiments are mostly in deterministic domains.
>
> - In the FetchSlide environment, the best-learned policy of CDP outperforms the baselines and PER, as shown in Table 1.
> Yes, we did not use the last set of parameters but used the best one encountered during training, as described in Section 4 “Experiments”: “After training, we use the best-learned policy as the final policy and test it in the environment. The testing results are the final mean success rates.“
>
> - Our implementation is based on “OpenAI Baselines”, which provides HER. We combined HER with PER in “OpenAI Baselines”.
> OpenAI Baselines link: https://github.com/openai/baselines
>
> - To improve the clarity of the paper, we move the exact set-up into the earlier section, Section 2.1 “Environments”.
> In this section, we also redefine the “state” based on your suggestions.
> We delete the “troublesome” sentence and also clarify what the goal actually is in Section 2.1.
> For more detail, please read the revised paper, Section 2.1.

---

### Official Review · AnonReviewer2 · 2018-11-03
**review for Curiosity-Driven Experience Prioritization via Density Estimation**

**Rating:** 6
**Confidence:** 3

**Review:**

The paper proposes a novel method for sampling examples for experience replay. It addresses the problem of having inbalanced data (in the experience buffer during training). The authors trained a density model and replay the trajectories that has a low density under the model.

Novelty:

The approach is related to prioritized experience replay, PER is computational expensive because of the TD error update, in comparison, CDR only updates trajectory density once per trajectory.

Clarity:

The paper seems to lack clarity on certain design/ architecture/ model decisions.  For example, the authors did not justify why VGMM model was chosen and how does it compare to other density estimators.  Also, I had to go through a large chunk of the paper before coming across the exact setup. I think the paper could benefit from having this in the earlier sections.


Other comments about the paper:

-  I do like the idea of the paper. It also seems that curiosity in this context seems to be very related to surprise? There are neuroscience evidence indicating that humans turns to remember (putting more weights) on events that are more surprising.

- The entire trajectory needs to be stored, so the memory wold grow with episode length. I could see this being an issue when episode length is too long.

---

> ### Author Response · Authors · 2018-11-17
> **To Reviewer**
>
> Thank you for the valuable feedback!
> We uploaded a revised version of the paper based on the comments.
>
> - To improve the clarity, we clarify why we chose to use V-GMM, among the three basic density estimation methods, including KDE, GMM, and V-GMM.  (in the revised version of the paper Section “2.3 Density Estimation Methods”)
> The reasons are the following:
> 1. GMM can be trained reasonably fast for RL agents. GMM is also much faster in inference compared to Kernel Density Estimate (KDE) (Rosenblatt, 1956).
> 2. Compared to GMM,  V-GMM has a natural tendency to set some mixing coefficients close to zero and generalizes better.
> 3. We only use a basic density estimation method, such as V-GMM, in our framework as a proof of concept for the idea “Curiosity-Driven Experience Prioritization via Density Estimation”. Other destiny estimation methods can also be applied in this framework.
>
> - We move the exact setup (Section “2.1 Environments” in the new version) in early sections to improve the clarity of the paper.
>
> - We are glad that you like the idea of the paper. Yes, indeed the curiosity mechanism in our context is related to surprise. The idea of our method is also related to neuroscience (Gruber et al., 2014).
>
> - Yes, the entire trajectories need to stored in the replay buffer and the memory size increases as the trajectory length increases. However, this is a general issue with off-policy RL methods which uses experience replay, such as DQN and DDPG. Our method CDP only uses the trajectories that are already in the memory, so CDP does not introduce additional memory usage.

---

### Author Response · Authors · 2018-11-17
**Revision**


- Move the exact set-up in early section (Section 2.1 Environments).
- Clarify the reason for using V-GMM in Section 2.3
- Define the “state” more formally in Section 2.1
- Add mathematical justification in Section 3.3 An Importance Sampling Perspective
- Other small fixes to improve the clarity of the paper

---

### Meta-Review · Area_Chair1 · 2018-12-14
**Insufficient clarity and detail, reviewer concerns not addressed.**

**Confidence:** 4
**Recommendation:** Reject

**Metareview:**

The manuscript describes a procedure for prioritizing the contents of an experience replay buffer in a UVFA setting based on a density model of the trajectory of the achieved goal states. A rank-based transformation of densities is used to stochastically prioritize the replay memory.

Reducing the sample complexity of RL is a worthy goal and reviewers found the overall approach is interesting, if somewhat arbitrary in the implementation details. Concerns were raised about clarity and justification, and the restriction of experiments to fully deterministic environments.

After personally reading the updated manuscript I found clarity to still be lacking. Statements like "... uses the ranking number (starting from zero) directly as the probability for sampling" -- this is not true (it is normalized, as confusingly laid out in equation 2 with the same symbol used for the unnormalized and normalized densities), and also implies that the least likely trajectory under the model is never sampled, which doesn't seem like a desirable property. Schaul's "prioritized experience replay" is cited for the choice of rank-based distribution, but the distribution employed in that work has rather different form. The related work section is also very poor given the existing body of literature on curiosity in a reinforcement learning context, and the new "importance sampling perspective" section serves little explicatory purpose given that an importance re-weighting is not performed.

Overall, I concur most strongly with AnonReviewer1 that more work is needed to motivate the method and prove its robustness applicability, as well as to polish the presentation.